# Prognostic value of acute National Institutes of Health Stroke Scale Items on disability: a registry study of first-ever stroke in the western part of Sweden

Guri Hagberg ,[1,2] Haakon Ihle-Hansen ,[3] Tamar Abzhandadze ,[2,4] Malin Reinholdsson,[2,4] Hege Ihle Hansen,[1,3] Katharina S Sunnerhagen [2,5]

For numbered affiliations see end of article.

**Correspondence to**
Professor Katharina S Sunnerhagen;
ks.sunnerhagen@neuro.gu.se

## ABSTRACT

**Objectives** We aimed to study how the individual items of the National Institutes of Health Stroke Scale (NIHSS) at admission predict functional independence 3 months post-stroke in patients with first-ever stroke.

**Setting** This registry-based study used data from two Swedish stroke registers (Riksstroke, the mandatory national quality register for stroke care in Sweden, and Väststroke, a local quality stroke register in Gothenburg).

**Participants** This study included patients with first-ever acute stroke admitted from November 2014 to August 2018, with available NIHSS at admission and modified Rankin Scale (mRS) at 3-month follow-up.

**Primary outcome** The primary outcome variable was mRS≤1 (defined as an excellent outcome) at 3-month follow-up.

**Results** We included 1471 patients, mean age was 72 (± 14.5) years, 48% were female, and 66% had mild strokes (NIHSS≤3). In adjusted binary logistic regression analysis, the NIHSS items impaired right motor arm and leg, and impairment in visual field, reduced the odds of an excellent outcome at 3 months ((OR 0.60 (95% CI 0.37 to 0.98), OR 0.60 (95% CI 0.37 to 0.97), and OR 0.65 (95% CI 0.45 to 0.94)). When exploring the effect size of associations between NIHSS items and mRS≤1 p, orientation, language and right leg motor had the largest yet small association.

**Conclusions** Stroke patients with scores on the NIHSS items right motor symptoms or visual field at admission are less likely to have an excellent outcome at 3 months. Clinicians should consider the NIHSS items affected, not only the total NIHSS score, both in treatment guidance and prognostics.

## STRENGTHS AND LIMITATIONS OF THIS STUDY

⇒ This study is based on registry data of high quality and high coverage, reflecting a real-life clinical setting, potentially minimising the risk of selection bias.
⇒ Registry data have limitations with missing data that might limit the generalizability.
⇒ To our knowledge, this study is the first to examine the prognostic merit of National Institutes of Health Stroke Scale individual items regardless of stroke severity on post-stroke functioning.

## INTRODUCTION

The National Institutes of Health Stroke Scale (NIHSS) includes 15 individual items measuring motor and sensory function, vision, level of consciousness, attention, neglect, language and speech production.[1][2] The NIHSS is the most widespread clinical scale used in patients presenting with acute stroke, and a high total NIHSS score is associated with poor outcomes in both haemorrhagic and ischaemic strokes.[3–5]

The total NIHSS score has several limitations as a predictor of later stroke outcomes. The 15 items are weighted similarly, scoring 0–42, but left brain strokes generally score 4 points higher on the NIHSS than right brain strokes of larger volume.[6] Total NIHSS scores at admission are inferior to 24 hours and discharge scores in predicting post-stroke functional outcomes, due to spontaneous improvement and the effect of acute interventions like thrombolysis or thrombectomy.[7][8] In strokes affecting posterior circulation, a low NIHSS might not predict a good functional outcome as the NIHSS is biased towards anterior circulation strokes.[9][10] In addition, in minor stroke, defined as NIHSS≤3p,[11] up to one-third of the patients are disabled or dead after 3 months.[12][13]

As the stroke field evolves, physicians tend to treat more small strokes with reperfusion therapy, although the published papers supporting this are lacking.[14] Current American Heart Association/American Stroke Association guidelines indicate uncertainty regarding the use of intravenous thrombolysis (IVT) in patients with low NIHSS scores

and non-disabling deficits,[15] and recent European guidelines suggest against IVT in acute minor non-disabling stroke.[16] Disabling strokes is not clearly defined in clinical practice, and excluding patients from acute treatment because their symptoms are considered too mild by the treating physician, might lead to unnecessary disability and deaths.[17 18] A suggested definition is that a disabling deficit would prevent the patient from performing basic activities of daily living or returning to work,[19] both clearly dependent on the patients' premorbid function and age.

In light of the widespread clinical use of NIHSS in the acute phase defining stroke severity and the total NIHSS scores limitations, knowing how the individual NIHSS items relate to functional outcomes after 3 months is essential in both treatment guidance and prognostics. Since few stroke registers report the full NIHSS with subitems, the evidence on the prognostic merit of NIHSS individual items regardless of stroke severity on post-stroke functioning is lacking. We aimed to study how the individual items of NIHSS at admission predicts functional independence, modified Rankin Scale (mRS)≤1, 3 months post-stroke in an unselected stroke population.

## METHODS
### Study design
This register-based study used data from two Swedish stroke registers (Riksstroke, the mandatory national quality register for stroke care in Sweden,[20] and Väststroke, a local quality stroke register in Gothenburg, collecting data from 2014 to 2019). The two registers provide different information about the same patients. The datasets were merged by a statistician at Riksstroke through personal identification numbers into one pseudonymized database.

### Study sample
This study includes all the patients≥ 18 years admitted with a first-ever acute stroke (ischaemic stroke (I63) or intracerebral haemorrhage (I61)) according to the International Classification of Diseases 10, to the Sahlgrenska University Hospital from November 2014 to August 2018, with available NIHSS items scores at admission and mRS at 3-month follow-up. Further, permanent institution residents were not invited to the follow-up, and together with patients with mRS=6 (dead) excluded from the study. The Sahlgrenska University Hospital consists of three sites, each with a comprehensive stroke unit, and provides emergency and basic care for 800 000 inhabitants in Gothenburg as well as specialized care for 1.8 million inhabitants in the west of Sweden.

### Procedure and study variables
Väststroke contains information on the NIHSS at admission, assessed by physicians. In cases with missing NIHSS in the Väststroke register, assessments were retrieved from medical records when possible.[21] In Sweden, patients who had a stroke are invited to a 3-month outpatient follow-up

visit for a medical review and assessment of functional status (mRS). The data on mRS were collected from the charts or by telephone from the nurses in a structured way.[22] All variables in this study, with the exception of mRS at 3 months, were collected at the index stay. In Riksstroke, the data were recorded by trained nurses working at the stroke units. Data retrieved from Riksstroke includes variables like sex, age, vascular risk factors, reperfusion treatment and discharge destination. The explanatory variables were NIHSS items, and each NIHSS item component on admission was dichotomized into; no symptom (score 0) or symptom(s) (score of 1 or more). The outcome variable was mRS; an excellent outcome was defined as mRS≤1.

### Statistics
Descriptive demographic and clinical measures are given as mean±SD, median and IQR or as numbers and percentages as appropriate. Categorical variables were compared with $\chi^2$ and Mann-Whitney U-test for continuous variables. Participants were divided into two groups based on mRS levels (mRS≤1, indicating an excellent outcome, or mRS>1, indicating slight to severe disability). In order to identify NIHSS items as predictors of an excellent outcome, logistic regression models were fitted and adjusted for the predefined variables such as age, sex, premorbid independency, stroke subtype and admission stroke severity (according to NIHSS; minor 0–3, mild to moderate 4–15; severe≥16). As an exploratory analysis, a second model stratified for the stroke subtypes, ischaemic stroke or haemorrhage, and adjusted for age, sex, premorbid independency, stroke severity and reperfusion therapy (in case of ischaemic stroke), were built (online supplemental table 1). The results are given in OR with a 95% CI. The effect size of the association between the excellent outcome and NIHSS items was investigated using phi coefficient; 0.1–0.29 (small effect), 0.30–0.49 (medium effect) and 0.5–1.0 (large effect).[23] All statistical tests were two-sided with alpha 5%. The analyses were performed using IBM SPSS Statistics V.28 software.

## RESULTS
Out of 3740 patients admitted with acute stroke, 1471 (39 %) patients had available data on NIHSS items at admission and mRS at 3-month follow-up; see the flow chart in figure 1. Among the 2269 not included patients, 374 (16.5 %) died during the hospital stay, and in general, they were older, with a mean age of 75±13.5 years, and had higher mean NIHSS (6.9±8) (online supplemental table 2).

Of 1471 included patients, 48% were female, the mean age was 72±14.5 years, and mean NIHSS 3.8±5.1 at stroke admission. Median (IQR) mRS at 3 months was 2.[2] Most (91%) of the strokes were ischaemic, and 66% were considered mild with NIHSS≤3. Facial palsy (in 31% of the stroke cases), followed by dysarthria (30%), was the most common scored NIHSS item. A significantly higher

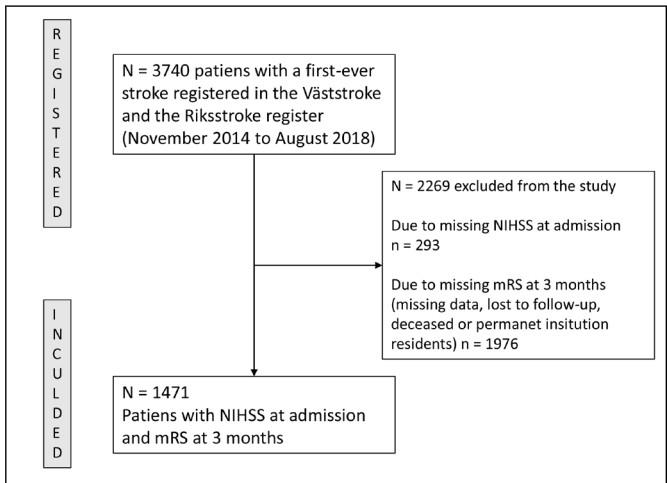

**Figure 1** Flow chart for study population. mRS, modified Rankin Scale; NIHSS, National Institutes of Health Stroke Scale.

proportion of patients with left-side than right-side motor symptoms (p<0.001). Patient characteristics and NIHSS items subscores are presented in table 1. The distribution of mRS at 3 months based on stroke severity at admission is shown in figure 2, where 26% of patients who had mild strokes had mRS≤1 at 3 months.

In binary logistic crude regression models, scores in any NIHSS subitems significantly reduced the odds of an excellent outcome (mRS≤1) at 3 months. After adjustments for the other independent variables (age, sex, premorbid independency, stroke subtype and stroke severity), scores on the right arm and leg motor symptoms and visual fields reduced the odds of an excellent outcome (mRS≤1) at 3 months, OR 0.60 (95% CI 0.37 to 0.98), OR 0.60 (95% CI 0.37 to 0.97), and OR 0.65 (95% CI 0.45 to 0.94), respectively (figure 3). When stratifying for ischaemic or haemorrhagic stroke, the results were comparable for ischaemic stroke to those for the entire cohort, but no NIHSS item or side predicted outcome in haemorrhagic stroke (online supplemental table 1).

When exploring the effect size of association between dichotomised NIHSS items and excellent outcome 3 months after stroke (mRS≤1), orientation, language and right leg motor had the largest, jet small, association (0.202, 0.183, and 0.182, p<0.001), respectively (table 2).

## DISCUSSION

In this cross-sectional retrospective and register-based study, patients with first-ever stroke and scores on the NIHSS items right motor symptoms or visual field at admission were less likely to have an excellent outcome at 3 months.

A possible explanation for the importance of the NIHSS right motor items for prognosis is the weighting of motor symptoms in the mRS and affection of the dominant hemisphere. The mRS attempts to measure functional independence, incorporating the WHO components of body

**Table 1** Characteristics of the study sample

| Total | 1471 |
|---|---|
| Age | |
| Mean (SD) | 71.8 (14.5) |
| Median (IQR) | 74 (82–63) |
| Sex (female) (%) | 658 (47.7) |
| Stroke type (%) | |
| Ischaemic | 1335 (90.8) |
| Haemorrhagic | 136 (9.2) |
| Reperfusion therapy (%) | 242 (16.5) |
| NIHSS | |
| Mean (SD) | 3.8 (5.1) |
| Median (IQR) | 2 (5-0) |
| Stroke severity (%) | |
| Mild (NIHSS≤3) | 965 (65.6) |
| Mild to moderate | 429 (29.2) |
| Severe (NIHSS>15) | 77 (5.2) |
| mRS 3 months | |
| Median (IQR) | 2 (3–1) |
| Excellent outcome (mRS≤1) | 703 (47.8) |
| Discharged to (%) | |
| Home | 1045 (71.0) |
| Premorbid independency* (%) | 1169 (79.5) |
| NIHSS subscore (% of 1471) | |
| 1a Consciousness | 90 (6.1) |
| 1b Orientation | 311 (21.1) |
| 1c Commands | 110 (7.5) |
| 2 Best Gaze | 169 (11.5) |
| 3 Visual field | 257 (17.5) |
| 4 Facial palsy | 454 (30.9) |
| 5 Motor arm | |
| Right | 147 (10.0) |
| Left | 211 (14.3) |
| 6 Motor leg | |
| Right | 159 (10.8) |
| Left | 213 (14.5) |
| 7 Ataxia | 206 (14.0) |
| 8 Sensory | 350 (23.8) |
| 9 Best language | 340 (23.1) |
| 10 Dysarthria | 438 (29.8) |
| 11 Neglect | 183 (12.4) |
| Cardiovascular risk/comorbidity (%) | |
| Atrial fibrillation | 310 (21.1) |
| Diabetes | 246 (16.7) |
| Smoking | 171 (11.6) |
| On statins | 300 (20.4) |
| Hypertension | 720 (48.9) |

*140 missing.
mRS, modified Rankin scale; NIHSS, National Institutes of Health Stroke Scale.

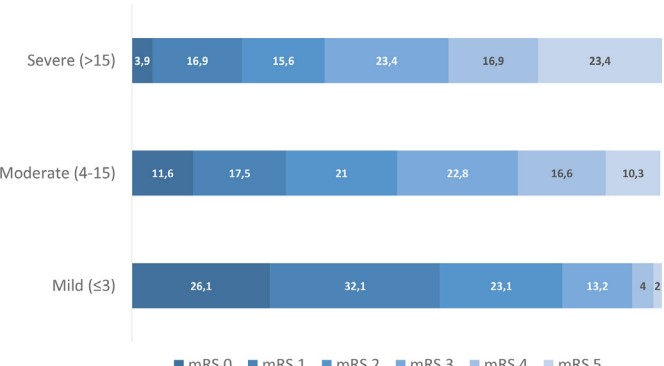

**Figure 2** Percentage of patients with different mRS at 3 months according to stroke severity at baseline (NIHSS; minor≤3, mild to moderate 4–15; severe>15). mRS, modified Rankin scale; NIHSS, National Institutes of Health Stroke Scale.

function, activity and participation, but domains such as cognition or language are not directly measured.[24 25] Abnormal motor items on the NIHSS in minor ischaemic strokes (NIHSS<5) has been linked to poor outcome in two other studies, but none of them looked at the side of lesions.[26 27] When assessing stroke outcomes by the side of involvement, the evidence is inconclusive, although one might think that affecting language and dominant hand function would negatively influence the outcome.[28 29] Several scale items require intact language function, and studies have shown that patients with left-sided stroke and total anterior circulation syndromes score four points higher on the NIHSS than right-sided stroke patients despite smaller stroke volumes on imaging.[6 30] The weighting of NIHSS items associated with left-side lesions makes the NIHSS less suitable for predicting outcomes for right-side strokes.[31] Our study, looking at dichotomized sub-items, supports the importance of dominant hemisphere affection.

**Figure 3** OR for excellent outcome (mRS≤1) with NIHSS items subscore in logistic regression (n=1471). *Unadjusted, †adjusted for age, gender, premorbid independency, stroke subtype and stroke severity. Each NIHSS item component on admission was dichotomised into; no symptoms (score 0) or symptoms (score of 1 or more). NIHSS, National Institutes of Health Stroke Scale.

The NIHSS item visual field reduces the odds of an excellent outcome in our study. Several studies have shown that visual field defects impact functional ability, quality of life, ability to participate in rehabilitation and poor long-term recovery.[32–37] Visual field defects could lead to loss of driver's license, automatically giving mRS>1, and linked to the risk of social isolation, depression and risk of requiring placement in long-term care.[38] The most common type of visual field loss is homonymous hemianopia, most common in strokes affecting posterior circulation, supplying optic radiations, occipital cortex and other brain areas involved in visual field.[39] NIHSS is generally less sensitive to strokes affecting posterior circulation, commonly causing vertigo and impaired balance,[40] and a low NIHSS does not reflect stroke severity.[9] In our cohort, 17.5% had visual field defects, fewer than in many other studies.[39 41] The screening nature of vision testing in the acute setting and many patients being unaware of their loss might explain this.[42] Our finding underpins the importance of visual field defects and testing in patients with otherwise low total NIHSS.

Our results could suggest that in patients with right motor symptoms or visual field defects, IVT should not be withheld due to low NIHSS alone, in line with previous work.[13] However, a recently published randomized clinical trial found that IVT was non-inferior to dual antiplatelet therapy regarding an excellent functional outcome at 90 days in minor non-disabling acute ischaemic stroke. They do not report on NIHSS items but included patients with NIHSS≤5, with less than or equal to one point on the subitems, including limb weakness or vision.[14] Both guidelines and study protocols describe minor or isolated non-disabling symptoms as reasons to withhold IVT, but no clear description exists. Suggested deficits are ataxia alone, sensory loss alone, dysarthria or mild aphasia alone, minimal weakness, isolated facial droop, or mild cortical hand.[16 19 43] As in our study, two other studies have shown that motor symptoms predict poor outcomes, but instead of vision, they report ataxia as a predictor.[9 26] However, this is not supported in our data.

In general, and in line with previous studies, the individual NIHSS items explained only a small portion of the variance in outcome.[26 44 45] Long-term follow-up with sensitive outcome measures is needed to make good prediction models, especially in minor strokes.

The strength of this register-based study is an unselected stroke cohort, with a complete NIHSS score including subitems and mRS at 3 months, collected in a real-life clinical setting, not excluding patients due to strict inclusion criteria. The overall coverage ratio in the Riksstroke register is 90%,[46] and taken together with the other factors mentioned above, the generalizability of our findings to countries with similar stroke populations and healthcare systems seems likely.

A limitation in our study is missing data, where only 39% had both available data on NIHSS items at admission and mRS at 3-month follow-up. However, as our drop-out analyses show that patients excluded had larger strokes

**Table 2** Exploring the strength of relationship between dichotomised NIHSS items and excellent outcome (mRS≤1)

| N (%) | Outcome according to mRS | | $\chi^2$ | Phi coefficient |
|---|---|---|---|---|
| | mRS>1 n (%) | mRS≤1 n (%) | | |
| 1a Consciousness | | | | |
| Absence | 690 (90.7) | 683 (97.3) | | |
| Symptoms | 71 (9.3) | 19 (2.7) | <0.001 | 0.138 |
| 1b Orientation (0 ref.) | | | | |
| Absence | 538 (70.7) | 611 (87.4) | | |
| Symptoms | 223 (29.3) | 88 (12.6) | <0.001 | 0.204 |
| 1c Commands | | | | |
| Absence | 671 (88.8) | 677 (96.4) | | |
| Symptoms | 85 (11.2) | 25 (3.6) | <0.001 | 0.145 |
| 2 Best Gaze | | | | |
| Absence | 608 (82.8) | 655 (93.8) | | |
| Symptoms | 126 (17.2) | 43 (6.2) | <0.001 | 0.171 |
| 3 Visual field | | | | |
| Absence | 545 (75.2) | 615 (88.9) | | |
| Symptoms | 180 (24.8) | 77 (11.1) | <0.001 | 0.178 |
| 4 Facial palsy | | | | |
| Absence | 462 (62.3) | 524 (75.1) | | |
| Symptoms | 280 (37.7) | 174 (24.9) | <0.001 | 0.138 |
| 5 Motor arm | | | | |
| Right | | | | |
| Absence | 643 (85.3) | 665 (94.9) | | |
| Symptoms | 111 (14.7) | 36 (5.1) | <0.001 | 0.159 |
| Left | | | | |
| Absence | 607 (80.2) | 639 (91.3) | | |
| Symptoms | 150 (19.8) | 61 (8.7) | <0.001 | 0.158 |
| 6 Motor leg | | | | |
| Right | | | | |
| Absence | 619 (83.4) | 664 (94.9) | | |
| Symptoms | 123 (16.5) | 36 (5.1) | <0.001 | 0.182 |
| Left | | | | |
| Absence | 595 (79.8) | 637 (91.1) | | |
| Symptoms | 151 (20.2) | 62 (8.9) | <0.001 | 0.160 |
| 7 Ataxia | | | | |
| Absence | 590 (81.6) | 619 (89.5) | | |
| Symptoms | 133 (18.4) | 73 (10.5) | <0.001 | 0.111 |
| 8 Sensory | | | | |
| Absence | 509 (69.3) | 556 (81.9) | | |
| Symptoms | 225 (30.7) | 125 (18.1) | <0.001 | 0.146 |
| 9 Best language | | | | |
| Absence | 500 (68.5) | 582 (84.1) | | |
| Symptoms | 230 (31.5) | 110 (15.9) | <0.001 | 0.183 |
| 10 Dysarthria | | | | |
| Absence | 448 (61.5) | 539 (77.4) | | |
| Symptoms | 281 (38.5) | 157 (22.6) | <0.001 | 0.173 |

**Table 2** Continued

| N (%) | Outcome according to mRS | | | |
| | mRS>1 n (%) | mRS≤1 n (%) | $\chi^2$ | Phi coefficient |
| --- | --- | --- | --- | --- |
| 11 Neglect | | | | |
| Absence | 561 (81.4) | 625 (91.9) | | |
| Symptoms | 128 (18.6) | 55 (8.1) | <0.001 | 0.154 |

Determining the effect size of the associations; effect size 0.1–0.29 (small), 0.30–0.49 (medium) and 0.5–1.0 (large), mRS, modified Rankin scale. Each NIHSS item component on admission was dichotomized into; no symptoms (score 0) or symptoms (score of 1 or more).
NIHSS, National Institutes of Health Stroke Scale.

and higher mortality, our conclusions can contribute to the decision-making process in the acute treatment of minor strokes where selection to and effect of IVT are less established. Another limitation is that we do not know the time from symptoms onset to NIHSS assessment, and studies have shown that NIHSS at 24 hours is better in prognostics, as many symptoms resolve during the first 24 hours.[7]

In conclusion, patients who had a stroke with scores on the NIHSS items right motor symptoms or visual field at admission are less likely to have an excellent outcome at 3 months. Clinicians should consider the NIHSS items affected, not only the total NIHSS score, both in treatment guidance and prognostics.

## Patient and public involvement

There were no patient or members of the public involved in the planning of this study.

**Author affiliations**
[1]Oslo Stroke Unit, Neurological Department, Oslo University Hospital, Ullevål, Oslo, Norway
[2]Institute of Neuroscience and Physiology, Section for Clinical Neuroscience and Rehabilitation, Gothenburg University, Goteborg, Sweden
[3]Bærum Hospital- Vestre Viken Hospital Trust, Department of Medicine, Drammen, Norway
[4]Department of Occupational Therapy and Physiotherapy, Sahlgrenska University Hospital, Goteborg, Sweden
[5]Neurocare, Sahlgrenska University Hospital, Goteborg, Sweden

**Acknowledgements** The authors thank the members of the Riksstroke Collaboration for data from the Swedish Stroke Register, Riksstroke (riksstroke.org), and the Väststroke Collaboration for data from the Väststroke Register. They also thank the participating patients and support staff for assistance with data acquisition.

**Contributors** GH: drafting of the manuscript. MR, TA: acquisition of data. All authors contributed to study concept and design, analysis and interpretation of data and critical revision of the manuscript for intellectual content. KS: guarantor.

**Funding** This work was supported by The Local Research and Development Board for Gothenburg and Södra Bohuslän, the Renée Eanders Foundation, the Swedish Stroke Association, the Swedish Heart-Lung Foundation, the Swedish Brain Foundation, Promobilia, and the Swedish Research Council VR 2017-00946. The Swedish state under the ALF agreement, an agreement between the Swedish government and county councils (ALFGBG-965653).

**Competing interests** None declared.

**Patient and public involvement** Patients and/or the public were not involved in the design, or conduct, or reporting, or dissemination plans of this research.

**Patient consent for publication** Not applicable.

**Ethics approval** The study was approved by the Regional Ethical Review Board in Gothenburg, Sweden on 4 May 2016, registration number 346-16, with an amendment approved on 14 September 2018, registration number T807-18. According to the Data Inspection Board in Sweden, data from quality registers are exempt from the general rule of patient consent (the Personal Data Act, Swedish law No. SFS 1998:204), since their purpose is to improve the quality of care, treatment, and rehabilitation which is of public interest. Furthermore, the patients were informed that their data were reported to the stroke registers and could be used for research. The patients may opt-out with withdrawal of their data at any time. The data were pseudonymised; thus, patients cannot be identified.

**Provenance and peer review** Not commissioned; externally peer reviewed.

**Data availability statement** Data are available upon reasonable request. Data from registries are subject to the Personal Data Act (Swedish law No. SFS 1998:204). Data may be available to researchers upon request, after review of secrecy (contact the author ks.sunnerhagen@neuro.gu.se). According to the Swedish regulation (epn.se/en/start/regulations/), the permission to use data can only be according to application and approval from the ethical board.

**ORCID iDs**
Guri Hagberg http://orcid.org/0000-0003-2822-7701
Haakon Ihle-Hansen http://orcid.org/0000-0001-7111-5970
Tamar Abzhandadze http://orcid.org/0000-0002-0069-6875
Katharina S Sunnerhagen http://orcid.org/0000-0002-5940-4400

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
