## [Reviewer comments · BMJ Open]

ARTICLE DETAILS

TITLE (PROVISIONAL)	The prognostic value of acute National Institutes of Health Stroke Scale Items on disability: A Registry Study of first-ever-stroke in the western part of Sweden
AUTHORS	Hagberg, Guri; Ihle-Hansen, Haakon; Abzhandadze, Tamar; Reinholdsson, Malin; Hansen, Hege; Sunnerhagen, Katharina

VERSION 1 – REVIEW

REVIEWER	Harring, Astrid Karina V. Oslo Metropolitan University, Department for prehospital education and research, Institute for Nursing and Health promotion
REVIEW RETURNED	26-Sep-2023

GENERAL COMMENTS	Thank you for the opportunity to review this research study. The study is well described and has an interesting aspect! While reading the manuscript in its current form, questions and comments came up - please find them in the attached PDF. In summary: My major concern is not the 5–10-year-old data this study is based on, but the age of the literature the study is seen and discussed in relation to. However, after a quick literature search, it seems plausible that it is due to the subject being so sparsely researched previously. If that is the case, I welcome to see that highlighted by the authors as it speaks to the novelty of this paper. There is also a bold claim for generalization that needs moderation. The uncommon ranking of "small, medium and strong" needs addressing and referencing to a framework for established limits/cut off's. There are minor adjustments needed such as the incongruent use of predict, explore, relates, etc (+ "explain" that is in the abstract introduction). I look very much forward to your response and to the see the revision!
--

REVIEWER	Li, Zixiao Beijing Tiantan Hospital
REVIEW RETURNED	10-Oct-2023

GENERAL COMMENTS	Thank you for the opportunity to review Dr. Hagberg's and colleagues' article. In this article, the authors present some
--

	important data on the prognostic value of acute National Institutes of Health Stroke Scale Items on disability. They aimed to study how the individual items of NIHSS at admission relate to functional independence three months post-stroke in a Registry Study of first-ever-stroke. Unsurprisingly, they found the NIHSS items impaired right motor arm and leg, and impairment in visual field, reduced the odds of an excellent outcome at three months. However, I have the following major concerns:  1. In the Study sample section and Figure 1. The procedures and criteria for patient inclusion are not very clear. 2. Only 39 % of patients had available data on NIHSS items at admission and mRS at three months follow-up. The data missing rate is too high. Is there a bias in patient enrollment? The clinical features of enrolled and excluded patients should be compared. 3. Page 4, line 29. What does the “p” mean? 4. 91 % of the strokes were ischemic. Are there differences between ischemic and hemorrhagic stroke? The authors should discuss the subgroup analysis results. 5. In Table 1, What does the percentage of the NIHSS sub score mean ? 6. It is not appropriate to use the Spearman rho to explore the strength of relationship between dichotomized NIHSS items and mRS. 7. Page 6, Line 37. “suggest that in patients with right motor symptoms or visual field defects, IVT should not be withhold due to low NIHSS alone,” Is there any definitive data to support the efficacy of IVT in this subset of patients? 8. The authors should consider the effect of treatment such as IVT on outcomes.
--	--

VERSION 1 – AUTHOR RESPONSE

Reviewer #1:

In summary: My major concern is not the 5–10-year-old data this study is based on, but the age of the literature the study is seen and discussed in relation to. However, after a quick literature search, it seems plausible that it is due to the subject being so sparsely researched previously. If that is the case, I welcome to see that highlighted by the authors as it speaks to the novelty of this paper.

Authors response: Thank you for pointing this out, we have included a bullet point in the ‘Strengths and limitations of this study’ section, and a sentence in the introduction.

-To our knowledge, we are the first to demonstrate the prognostic merit of NIHSS individual items regardless of stroke severity on post-stroke functioning.

- Since few stroke registers report the full NIHSS with sub-items, the evidence on the prognostic merit of NIHSS individual items regardless of stroke severity on post-stroke functioning is lacking.

There is also a bold claim for generalization that needs moderation.

Authors response: We agree and have moderated our statement in the paper.

-The overall coverage ratio in the Riksstroke register is 90 % (ref), and taken together with the other factors mentioned above, the generalizability of our findings to countries with similar stroke population and healthcare system seems likely.

The uncommon ranking of "small, medium and strong" needs addressing and referencing to a framework for established limits/cut off's.

Authors response: Thank you, reviewer2 has pointed out that we did not use the right statistic test in this setting. Accordingly, we have consulted a statistician and changed our methods. In the revised manuscript we use phi coefficient to assess the effect size of the association. We have changed the abstract, method/statistics, result part and Table 2 in line with these adjustments. Further, included new reference to the statistics (Psychol Bull.1992 Jul;112(1):155-9. doi: 10.1037//0033-2909.112.1.155.).

There are minor adjustments needed such as the incongruent use of predict, explore, relates, etc (+ "explain" that is in the abstract introduction).

Authors response: Thank you for this comment, we totally agree, and have changed to predict in relevant places in the manuscript accordingly.

Other comment and suggestion for minor changes to the manuscript is given as comments in the PDF peer review file. We have changed the manuscript in line with reviewer 1 suggestions.

Page 3 line 8; Change to something like; can indicate....

Authors response: Thank you, changed in the manuscript.

Page 3 line 19; place inside the parentheses

Authors response: Thank you, changed in the manuscript.

Page 4 line 37; flow issue..

Authors response: Thank you, nice suggestion, changed in the manuscript.

Page 4 line 43-44; amend to predicts/Indicates...

Authors response: Thank you for this comment, we agree, and have changed to predict in relevant places in the manuscript, accordingly, as mentioned above.

Page 5 line 22-23; possibly place in study sample..

Authors response: Thank you, moved to study sample.

Page 5 line 45-46; change to order of the supplementary tables.

Authors response: Thank you for pointing this out, we have changed the order.

Page 5 line 49-50; does not seem right, revise...

Authors response: We agree and have change the method and manuscript as described.

Page 6 line 23: Excellent. I applaud the authors for their transparency.

Authors response: Thank you for this nice comment.

Page 6 line 26; Change to: Most (91 %)...

Authors response: Thank you, agree, changed in the manuscript.

Page 6 line 30-31; Is this unique or known from literature?

Authors response: This depends on the study population, and is not unique, but in line with some other studies, others not.

Page 6 line 51; here you use the wording "low"..

Authors response: Thank you, changed to small.

Page 7 line 44; what are your finding regarding ataxia.

Authors response: Thank you, changed the sentence in the manuscript, trying to make it clearer; As in our study, two other studies have shown that motor symptoms predict poor outcome, but instead of vision they report ataxia as a predictor (9, 24), not supported in our data.

Page 8 line 3; not only in 40 missing due to this..... and page 8 line 10; great. Highlight this more..

Authors response: We have restructured the limitation paragraph; A limitation in our study is missing data, where only 39 % had both available data on NIHSS items at admission and mRS at three months follow-up. However, as our drop-out analyses show that patients excluded had larger strokes and higher mortality, our conclusions can contribute to the decision-making process in acute treatment of minor stroke were selection to and effect of IVT are less established.

Page 8 line 24; T. (To keep consistent..)

Authors response: Thank you, changed in the manuscript.

Page 13 line 10; What was the n for, mild, moderate and severe? And I assume it is in %? Please clarify and supply a legend.

Authors response: Thank you, we have included an extended legend; Percentage of patients with different mRS according to stroke severity (NIHSS; minor ≤ 3 , mild to moderate 4-15; severe >15). NIHSS=National Institutes of Health Stroke Scale; mRS=modified Rankin scale.

Page 13 line 13; I assume all 1471 are included here?

Authors response: Thank you for that comment. We have placed the N in the figure title.

Following comments needs to be discussed in more detail:

Comment page 6 line 43; well. 91% had a ischemic stroke so they are most of the entire cohort.. either compare the two stratified groups with each other, or possibly delete this part.

Authors response: In the stratified analysis in Supplementary table 2, the confidence intervals overlap, so there are no significant differences between the groups. However, we do not find this highly relevant for the main aim in general, and as you point out, very few cases are in the haemorrhage group, and it could be removed. However, since reviewer2 requested this analysis, and after consulting a statistician, we suggest keeping it in Supplementary.

Comment line 57; ref. for the overall coverage ratio of the Riksstroke register.

Authors response: we have added a new ref. Söderholm A, Stegmayr B, Glader EL, Asplund K. Validation of Hospital Performance Measures of Acute Stroke Care Quality. Riksstroke, the Swedish Stroke Register. Neuroepidemiology. 2016;46(4):229-34.

Comment line 11 in References; Limitations; I suspect the last literature review was in 2018 and need re-doing, as just two references are newer. Has new insight occurred in the meantime? Is there no re-validation of these studies from the late 80ie-early 2000?

Authors response: we understand that our reference list might seem out of date, but the truth is that few have investigated this research question, especially in recent years – as now highlighted in our introduction. As the stroke field evolves, physicians tend to treat more small strokes with reperfusion therapy, although the published papers supporting this is lacking – reflecting the uncertainty in guidelines. This is also why we wanted to explore this although the NIHSS and most studies on IVT are old.

That said, we appreciate the careful review, and recognize that the latest update of the AHA guideline was not included in the reference list by mistake - we have updated the AHA/ASA guideline in to the Update from 2018-2019 (Dec;50(12):e344-e418. doi: 10.1161/STR.000000000000211. Epub 2019 Oct 30). We have also added a recent observational study looking at outcomes in minor stroke patients treated with intravenous thrombolysis, (2023 Aug;29(8):2308-2317. doi: 10.1111/cns.14164. Epub 2023 Mar 21.) and a newly published RCT (based on reviewer2 comments) (JAMA. 2023 Jun 27;329(24):2135-2144. doi: 10.1001/jama.2023.7827.)– to make it clearer that this still is an ongoing unsolved issue in the stroke field.

Page 11 line 16-17; Only 16% received thrombolysis! It's understandable that we don't know the time form symptoms to the first NIHSS assessment, but they must include that only 16% received thrombolysis in the limitations.

Authors response: Only 16 % received thrombolysis in our cohort, but that is in line with the complete Riksstroke register, where 17 % received thrombolysis, 3 % with thrombolysis and thrombectomy, and 4 % thrombectomy only in 2021. (https://www.riksstroke.org/wp-content/uploads/2022/11/Arsrapport-2021-engelsk-sammanfattning_final.pdf). In Norway, 20 % received thrombolysis, 5.9 % thrombectomy in 2021.

(<https://stolav.no/Documents/%c3%85rsrapport%20Norsk%20hjerneslagregister%202021.pdf>)

As we do not compare patients treated with reperfusion therapy and those not, and our regression models are adjusted for reperfusion therapy, it is not clear why we should mention this as a limitation.

Reviewer #2:

1. In the Study sample section and Figure 1. The procedures and criteria for patient inclusion are not very clear.

Authors response:

Thank you for this important comment. We have revised the study design and sample paragraph and included a new flow-chart.

2. Only 39 % of patients had available data on NIHSS items at admission and mRS at three months follow-up. The data missing rate is too high. Is there a bias in patient enrollment? The clinical features of enrolled and excluded patients should be compared.

Authors response: We agree, unfortunately this was the case, and we have already included the drop-out analysis in supplementary table 2. As mentioned to reviewer1, we have restructured the limitation paragraph; A limitation in our study is missing data, where only 39 % had both available data on NIHSS items at admission and mRS at three months follow-up. However, as our drop-out analyses show that patients excluded had larger strokes and higher mortality, our conclusions can contribute in the decision making process in acute treatment of minor stroke where selection to and effect of IVT are less established.

3. Page 4, line 29. What does the “p” mean?

Authors response: Thank you, we have removed the p.

4. 91 % of the strokes were ischemic. Are there differences between ischemic and hemorrhagic

stroke? The authors should discuss the subgroup analysis results.

Authors response: We have already included a stratified analysis in Supplementary table 2, and as you can see, the confidence intervals overlap, so there are no significant differences between the groups. However, we do not find this is highly relevant for the main aim in general, and as reviewer 1 points out, very few cases are in the haemorrhage group, and suggest it could be removed. However, since you requested this, we will still include the data in Supplementary, and we have made minor adjustment in the manuscript.

5. In Table 1, What does the percentage of the NIHSS sub score mean ?

Authors response: This means the proportion for the total included study population. 90 out of 1471 scored on consciousness, 6.1 %. We have included the total n in the heading in this section for clarity.

6. It is not appropriate to use the Spearman rho to explore the strength of relationship between dichotomized NIHSS items and mRS.

Authors response: Thank you for pointing this out. After reading your comment we have consulted a statistician and made changes the statistic according to his recommendations. In the revised manuscript we use phi coefficient to assess the effect size of the association. We have changed the abstract, method/statistics, result part and Table 2, accordingly. Further, included new reference to the statistics (Psychol Bull.1992 Jul;112(1):155-9. doi: 10.1037//0033-2909.112.1.155.).

7. Page 6, Line 37. "suggest that in patients with right motor symptoms or visual field defects, IVT should not be withhold due to low NIHSS alone, " Is there any definitive data to support the efficacy of IVT in this subset of patients?

Authors response: Thank you for this important comment. No, to our knowledge this data is lacking. A recent published observational study has not looked at NIHSS items, but only total NIHSS score. This article is added based on reviewer 1 comments- (Outcomes in minor stroke patients treated with intravenous thrombolysis, (2023 Aug;29(8):2308-2317. doi: 10.1111/cns.14164. Epub 2023 Mar 21.) However, since we wrote this manuscript, a RCT found that IVT was noninferior to dual anti platelet therapy regarding excellent functional outcome at 90 days in minor nondisabling acute ischemic stroke. They do not report subgroup analysis looking at visual field or right side weakness, but we have moderated our statement, and added the article to the discussion. (JAMA.2023 Jun 27;329(24):2135-2144. doi: 10.1001/jama.2023.7827.)

8. The authors should consider the effect of treatment such as IVT on outcomes.

Authors response: We agree and have included this in the discussion based on your prior comments.

Thanks again for many insightful comments.

VERSION 2 – REVIEW

REVIEWER	Harring, Astrid Karina V. Oslo Metropolitan University, Department for prehospital education and research, Institute for Nursing and Health promotion
REVIEW RETURNED	09-Nov-2023

GENERAL COMMENTS	Thank you to the authors for revising the manuscript. I appreciate the thoughtful and detailed responses to the comments. The paper has been significantly improved, and you have consulted a statistician and brought in some new references (such as #14, #23 etc). I only have one question and one minor suggestion. One question for the authors/editor: A new name is linked to many of the "track changes", please make sure anyone who deserves recognition is listed either as a co-author or in the Acknowledgments, depending on their involvement. One last minor suggestion:but instead of vision they report ataxia as a predictor (9, 26), not supported in our data COMMENT: flow issue, add something like "(9, 26), however this is..." Thank you for the opportunity to review this manuscript and I recommend going forward with publication without further reviews. I hope you continue to pursue this novel topic and encourage you to highlight the research gap and your findings at conferences and in guidelines etc.
--

VERSION 2 – AUTHOR RESPONSE

Reviewer #1:

-A new name is linked to many of the "track changes", please make sure anyone who deserves recognition is listed either as a co-author or in the Acknowledgments, depending on their involvement.

Authors response: Thank you for pointing this out. Morten Ødegaard is Guri Hagbergs husband and by default his name came up during the revision as they share computer at home. He did not have anything to do with this article. We apologize for the inconvenience.

-One last minor suggestion:

....but instead of vision they report ataxia as a predictor (9, 26), not supported in our data COMMENT: flow issue, add something like "(9, 26), however this is..."

Authors response: we agree and have made changes in the manuscript accordingly.

-Thank you for the opportunity to review this manuscript and I recommend going forward with publication without further reviews. I hope you continue to pursue this novel topic and encourage you to highlight the research gap and your findings at conferences and in guidelines etc.

Authors response: thank you for this comment, we agree, and will use any opportunity in the future to further investigate and discuss this important topic.